# Involvement of Mitochondrial Mechanisms in the Cytostatic Effect of Desethylamiodarone in B16F10 Melanoma Cells

**DOI:** 10.3390/ijms21197346

**Published:** 2020-10-05

**Authors:** Fadi H. J. Ramadan, Aliz Szabo, Dominika Kovacs, Aniko Takatsy, Rita Bognar, Ferenc Gallyas, Zita Bognar

**Affiliations:** 1Department of Biochemistry and Medical Chemistry, University of Pécs Medical School, 7624 Pecs, Hungary; fadi.ramadan@aok.pte.hu (F.H.J.R.); aliz.szabo@aok.pte.hu (A.S.); dominika.kovacs@aok.pte.hu (D.K.); Aniko.Takatsy@aok.pte.hu (A.T.); rita.bognar@aok.pte.hu (R.B.); zita.bognar@aok.pte.hu (Z.B.); 2Szentagothai Research Centre, University of Pécs, 7624 Pecs, Hungary; 3HAS-UP Nuclear-Mitochondrial Interactions Research Group, 1245 Budapest, Hungary

**Keywords:** amiodarone, apoptosis, metastatic melanoma, mitochondrial fusion-fission, mPT, Bad, Akt, Aif

## Abstract

Previously, we showed that desethylamiodarone (DEA), a major metabolite of the widely used antiarrhythmic drug amiodarone, has direct mitochondrial effects. We hypothesized that these effects account for its observed cytotoxic properties and ability to limit in vivo metastasis. Accordingly, we examined DEA’s rapid (3–12 h) cytotoxicity and its early (3–6 h) effects on various mitochondrial processes in B16F10 melanoma cells. DEA did not affect cellular oxygen radical formation, as determined using two fluorescent dyes. However, it did decrease the mitochondrial transmembrane potential, as assessed by JC-1 dye and fluorescence microscopy. It also induced mitochondrial fragmentation, as visualized by confocal fluorescence microscopy. DEA decreased maximal respiration, ATP production, coupling efficiency, glycolysis, and non-mitochondrial oxygen consumption measured by a Seahorse cellular energy metabolism analyzer. In addition, it induced a cyclosporine A–independent mitochondrial permeability transition, as determined by Co^2+^-mediated calcein fluorescence quenching measured using a high-content imaging system. DEA also caused outer mitochondrial membrane permeabilization, as assessed by the immunoblot analysis of cytochrome C, apoptosis inducing factor, Akt, phospho-Akt, Bad, and phospho-Bad. All of these data supported our initial hypothesis.

## 1. Introduction

Mitochondrial processes are indispensable for most eukaryotic cells. As a metabolic compartment, mitochondria produce ~90% of cellular ATP and represent a hub in the catabolism and synthesis of essential intermediates and macromolecule precursors for cell growth and proliferation [1]. In addition, mitochondria participate in the maintenance of Ca^2+^ homeostasis and redox balance, as well as the regulation of cell death [2]. Notably, mitochondria can regulate their functions according to the metabolic demands of the cell [3], and these functions are characteristically altered in cancer cells [4]. Accordingly, cancer is regarded as a mitochondrial metabolic disease [4,5,6,7], and intensive research efforts have been devoted to the identification of novel mitochondria-targeted therapeutic strategies [3,8].

Desethylamiodarone (DEA), the major metabolite of amiodarone (AM), accumulates rapidly in the extracardiac tissues of patients during AM antiarrhythmic therapy [9]. Although AM is one of the most frequently prescribed antiarrhythmics in the United States and is widely used in various life-threatening ventricular tachyarrhythmias worldwide, AM therapy is often limited by the toxic side effects of both the parent molecule and DEA itself [10]. These side effects include thyroid, hepatic, pulmonary, cardiac, psychiatric, ocular, dermatologic, hematological, and neuromuscular symptoms [10], which in most cases appear when the AM plasma concentration exceeds the recommended therapeutic value of 5.7 μM [11]. The tissue concentrations of AM and DEA can be 100–1000 times higher than the corresponding plasma concentration [11]. The accumulation of DEA in the myocardium, as well as in lung, liver, thyroid gland, pancreas, and skin, but not in adipose tissue, exceeds that of the parent molecule [9,12,13], and its mean elimination half-life is about 40 days [14]. Based on the tissue accumulation properties and toxic effects of DEA, we previously proposed that the compound has a potential use in cancer therapy and provided experimental evidence for its cytostatic and metastasis-limiting properties in the bladder, cervix, and melanoma cell lines [15,16,17].

In previous studies in isolated liver and heart mitochondria, we found that DEA at concentrations of 10–30 µM inhibits the mitochondrial respiratory chain, collapses the mitochondrial membrane potential (ΔΨ_m_), and induces a mitochondrial permeability transition (mPT) [18]. We concluded that these effects could account for the toxic side effects of AM [18]. In light of the dependence of cancer cells on optimized mitochondrial metabolism, these effects of DEA may contribute to its observed cytotoxic and metastasis-limiting properties. To explore this possibility, in this study we examined the effects of DEA treatment on mitochondrial processes in B16F10 melanoma cells.

## 2. Results

### 2.1. Effect of DEA on Cellular Reactive Oxygen Species (ROS) Production

Cancer cells and tissues exist under persistent oxidative stress, which affects their survival and metastatic properties [19]. Therefore, we were interested in whether DEA has any effect on ROS formation in the B16F10 melanoma line. To resolve this issue, we treated the cells with 5 and 10 µM of DEA in the presence of the non-fluorescent reduced derivatives of fluorescent redox dyes, and registered the increase in fluorescence resulting from the oxidizing effect of cellular ROS. We used two different fluorescent redox dyes to asses ROS formation separately in the aqueous and membranous compartments. Additionally, we used MitoSOX^TM^ Red, a redox dye of red fluorescence targeted to the mitochondria, where it is oxidized selectively by superoxide [20]. Taxol, which generates ROS at low micromolar concentrations, was used as a positive control [21]. In agreement with our previous results obtained in isolated liver and heart mitochondria [18], we did not detect significantly elevated ROS levels in either the aqueous or membranous compartments of B16F10 melanoma cells (Figure 1). However, we found that 10 µM of DEA did induce mitochondrial superoxide formation, which was much smaller in extent than the one caused by 5 µM of Taxol (Figure 1).

### 2.2. Effect of DEA on the Mitochondrial Membrane Potential (ΔΨ_m_)

ΔΨ_m_ is a significant factor in ATP synthesis and has a number of non-energetic functions essential for cell survival [22]. Accordingly, we were interested in how DEA treatment affects ΔΨ_m_ in B16F10 melanoma cells. For this purpose, we used a membrane potential-dependent fluorescent dye, JC-1, which accumulates in the mitochondria due to its positive charge. When ΔΨ_m_ is normal, the dye forms J-aggregates that emit red fluorescence upon excitation. Depolarization decreases the abundance of the dye in the mitochondria; consequently, the aggregates fall apart, and the monomer dye emits a green fluorescence when excited at 490 nm. When ΔΨ_m_ dissipates completely, the dye is not retained in the mitochondria, manifesting as a loss of fluorescence. Within 3 h, treatment with > 5 µM of DEA markedly depolarized the mitochondria in intact B16F10 melanoma cells (Figure 2).

### 2.3. Effect of DEA on Mitochondrial Fragmentation in Intact B16F10 Melanoma Cells

Mitochondrial fusion requires healthy ΔΨ_m_, which forms the basis of mitochondrial quality control [22]. Therefore, compromised ΔΨ_m_ results in mitochondrial fragmentation, raising the possibility that DEA treatment causes such an effect. To test this possibility, we performed confocal fluorescence microscopy on DEA-treated B16F10 melanoma cells transiently transfected with mitochondria-targeted red fluorescent protein (mtRFP) expressing vector [23]. Under these conditions, the intensity of the mitochondrially localized fluorescence did not depend on ΔΨ_m_. Treatment for 3 h with 10 µM of DEA resulted in mitochondrial fragmentation comparable to that caused by 25 µM of cisplatin (positive control [24]) (Figure 3). Although treatment with 5 µM of DEA tended to increase the mitochondrial fragmentation, the difference from the control did not reach the level of statistical significance (Figure 3).

### 2.4. Effect of DEA on the Energy Metabolism of B16F10 Melanoma Cells

Given that the DEA treatment compromised ΔΨ_m_, a major determinant of ATP synthesis [22], we next sought to assess DEA’s effect on energy metabolism of B16F10 melanoma cells. We used the Seahorse XF Cell Mito Stress Test to monitor the cellular oxygen consumption rate (OCR), an indicator of mitochondrial respiration, and the extracellular acidification rate (ECAR), an indicator of aerobic glycolysis in live B16F10 melanoma cells. We treated cells with 5 or 10 µM of DEA for 3 h, and then monitored the OCR and ECAR for 75 min. After registering basal respiration (Figure 4a-1) for 15 min, we added oligomycin, an inhibitor of F_o_F_1_ ATP synthase, to assess the ATP production (Figure 4a-4). After 20 min of recording, we added carbonyl cyanide 4-(trifluoromethoxy) phenylhydrazone (FCCP), which uncouples respiration and ATP synthesis, to measure maximal respiration (Figure 4a-3). After a further 20 min of recording, we inhibited mitochondrial respiration by adding the Complex I inhibitor rotenone and the Complex III inhibitor antimycin A to determine the proton leak and non-mitochondrial oxygen consumption (Figure 4a-2, Figure 4a-5). Simultaneously, we also monitored ECAR (Figure 4b). From the original recordings, the instrument output multiple parameters of cellular energy metabolism (Figure 4c–k). The DEA treatment did not significantly affect basal respiration (Figure 4c) or proton leak (Figure 4d), although 10 µM of DEA tended to decrease the former and increase the latter. By contrast, 10 µM of DEA did suppress the maximal respiration (Figure 4e), ATP production (Figure 4f), and coupling efficiency (Figure 4g); the last of these indicates how tightly respiration is coupled to ATP synthesis. DEA decreased the non-mitochondrial oxygen consumption (Figure 4h), and spare respiratory capacity, expressed either as the difference (Figure 4i) or ratio (Figure 4j) of maximal and basal respiration, in a concentration-dependent manner. Although it did not affect the basal fermentative ATP synthesis, DEA decreased lactate accumulation after oligomycin administration in a concentration-dependent manner (Figure 4b–k), indicating that the drug interfered with the glycolytic machinery as well as the mitochondrial respiratory chain.

### 2.5. Effect of DEA on mPT in Intact B16F10 Melanoma Cells

According to our previous results, DEA induced mPT in isolated liver and heart mitochondria that was resistant to cyclosporine A, a specific inhibitor of mPIT induced by elevated Ca^2+^ [26]. We were interested in determining whether the drug has the same effect in melanoma cells. Therefore, we adapted the method of Petronilli et al. [27] for measuring mPT in intact live cells using a 96-well automated high-content fluorescence imaging system. The method is based on the quenching of calcein fluorescence by Co^2+^. Non-fluorescent acetoxymethylcalcein is taken up by the cells, and is converted intracellularly to fluorescent calcein by non-specific esterases. Co^2+^, the cellular uptake of which is facilitated by the Ca^2+^ ionophore A23187, quenches the cytoplasmic calcein fluorescence; however, it cannot enter intact mitochondria, resulting in exclusively mitochondrial calcein fluorescence. When the mPT pore opens, Co^2+^ is free to enter the mitochondria and quench the calcein fluorescence there as well. Accordingly, we monitored the calcein fluorescence of melanoma cells treated with 5 or 10 µM of DEA or 1.5 mM of CaCl_2_ (positive control [27]) in the presence of acetoxymethylcalcein, CoCl_2_, and A23187 with or without CsA for 3 h. In complete agreement with our previous results obtained in isolated liver and heart mitochondria [18], 10 µM of DEA induced mPT that was CsA-independent (Figure 5). As expected, elevated Ca^2+^, the cellular uptake of which is facilitated by A23187, caused a massive mPT that was fully CsA-dependent (Figure 5).

### 2.6. Effect of DEA on Outer Mitochondrial Membrane (OMM) Permeabilization

OMM permeabilization can lead to apoptotic death [28]; therefore, we studied the effect of DEA on OMM integrity. When the OMM is intact, pro-apoptotic intermembrane proteins such as cytochrome C (Cyt c) and apoptosis-inducing factor (Aif) are retained in the mitochondria. Anti-apoptotic members of the B-cell lymphoma 2 (Bcl2) family form heterodimers with pro-apoptotic members, thereby inactivating them. When the Bcl2-associated agonist of cell death (Bad) is dephosphorylated, it binds to anti-apoptotic Bcl2 family members, releasing the pro-apoptotic ones, which in turn dimerize with each other, translocate to the OMM, and permeabilize it. This leads to a release of the pro-apoptotic intermembrane proteins, eventually resulting in apoptotic cell death [28]. A major mechanism of protection against apoptosis is the phosphorylation of Bad by the cytoprotective kinase Akt [28]. Optic atrophy1 (Opa1) is an inner mitochondrial membrane (IMM)-associated large GTPase protein that is present in the mitochondria only [29]. Stimuli leading to OMM permeabilization and Cyt c release results also in release of Opa1 to the cytosol [30]. Accordingly, we assessed DEA’s effect on OMM integrity by determining its effect on the expression, localization, and activation of Cyt c, Opa1, Aif, Bad, and Akt. To this end, we prepared whole-cell homogenate, and in parallel nuclear and cytoplasmic fractions from B16F10 melanoma cells treated with different concentrations of DEA for 6 h, and then subjected them to immunoblot analysis. To determine the phosphorylation states of Akt and Bad, we used phosphorylation-specific primary antibodies. At a concentration of 10 µM, DEA increased the steady-state level of Bad and decreased Bad phosphorylation in a concentration-dependent manner (Figure 6). Both of these effects shift the balance in the pro-apoptotic direction. Accordingly, DEA induces the release of Cyt c and Opa1 into the cytosol and nuclear translocation of Aif (Figure 6). In addition, 10 µM of DEA decreased Akt phosphorylation without affecting the steady-state level of the enzyme (Figure 6). This latter effect of DEA was fully consistent with its effects on the other proteins studied. Together, these data indicated that DEA caused OMM permeabilization.

### 2.7. Effect of DEA and Akt Inhibitors on the Viability of B16F10 Melanoma Cells

Because of the importance of mitochondrial processes in cancer [5], we were interested in determining whether the observed mitochondrial effects of DEA manifested as viability changes in B16F10 melanoma cells. To this end, we treated the cells for 3–12 h with 5 or 10 µM of DEA before determining their viability using the sulforhodamine B (SRB) assay. This assay is considered the most suitable for assessing the toxicity of substances in cultured cells, especially when the toxicity affects the mitochondria [31]. In complete agreement with our previous results [17], DEA decreased the viability of B16F10 melanoma cells in a time- and concentration-dependent manner, although at 5 µM and for up to 6 h of incubation, this effect did not reach statistical significance (Figure 7).

## 3. Discussion

Mitochondria contribute in multiple ways to carcinogenesis, tumor survival, and metastasis. The respiratory chain, situated in the inner mitochondrial membrane (IMM), is responsible for the majority of the ATP production under physiological conditions; moreover, it is a massive source of ROS formation in various maladies, including cancer [32,33]. Different ROS-associated signaling pathways have been proposed as oncologic therapeutic targets, particularly the nuclear factor erythroid 2-related factor 2 (Nrf2)–regulated anti-oxidant defense system [34,35]. A number of antineoplastic drugs and substances that inhibit the mitochondrial respiratory chain increase cellular ROS production [21,36,37,38]. Using a mitochondria-targeted fluorescent redox dye, we found that DEA at a concentration of 10 µM but not at 5 µM elicited mitochondrial superoxide formation (Figure 1). However, we failed to observe significantly elevated ROS levels either in the aqueous or membranous cellular compartments (Figure 1). The latter results are in accordance with our previous results obtained in isolated liver and heart mitochondria [18]. It is not clear whether this discrepancy was due to technical issues, or simply if the extent of mitochondrial superoxide production was insufficient to elevate the overall cellular ROS level. Indeed, oxidases—most notably, members of the NADPH oxidase family—are the major source of cellular ROS production besides the mitochondrial respiratory chain. Additionally, the latter is tightly controlled, and the superoxide produced by it is readily neutralized by antioxidant enzymes [39]. However, it is worth noting that the DEA-induced mitochondrial superoxide production was less than 20% of the one caused by 5 µM of Taxol (Figure 1), and the Taxol-induced cellular ROS production was detectable by conventional fluorescent redox dyes [21], not only by MitoSOX^TM^ Red.

In addition to its role in providing the driving force for ATP synthesis, ΔΨ_m_ plays essential roles in the transport of mitochondrial proteins encoded in the nucleus [40], as well as in cations such as K^+^, Ca^2+^, and Mg^2+^ [22]; ROS generation [41] and mitochondrial network dynamics [42]; and the regulation of cell death via the release of pro-apoptotic intermembrane proteins [26,43,44]. The maintenance of ΔΨ_m_ is so important for cell survival that in ischemic situations, the F_o_F_1_ ATPase can operate in reverse mode, consuming rather than synthesizing ATP to rescue the cell. Under such conditions, the substrate-level phosphorylation of non-glucose substrates supplies the ATP to be cleaved by the F_o_F_1_ ATPase, providing a narrow survival window [45,46,47]. Cancer cells, especially those in solid tumors, exist in a permanent state of partial ischemia, to which their cytoplasmic and mitochondrial metabolic pathways must adapt [3,48]. Therefore, drugs that interfere with tumor cell metabolism could be of therapeutic value [3]. DEA is an example of such a candidate drug: it decreased ΔΨ_m_ in a concentration-dependent manner in B16F10 melanoma cells, and a 3 h treatment with 10 µM of DEA massively decreased ΔΨ_m_ (Figure 2). This finding was in accord with our previous results obtained in isolated liver and rat mitochondria [18].

Mitochondrial network dynamics play important roles in mitochondrial biogenesis, the satisfaction of cellular energy and metabolic demands, retrograde signaling, and mitochondrial quality control [42,49]. To fulfill these tasks, continuous mitochondrial fusion and fission processes must be maintained in balance, and are mainly regulated by intracellular signaling [49]. However, ΔΨ_m_ plays a decisive role in this balance, as below a certain ΔΨ_m_ threshold the fusion process cannot take place [42,49]. Accordingly, excessive fission is a common feature of many tumors [50,51]. Although mitochondrial fission does not necessarily lead to apoptotic cell death, fragmented mitochondria are more susceptible to damage and are more apt to be eliminated by mitophagy, which forms the basis of mitochondrial quality control [52]. Excessive fission can lead to a significantly reduced mitochondrial DNA copy number, as observed in various malignancies including astrocytomas; prostate cancers; and breast, colon, and hepatocellular carcinomas [42]. Decreases in the mitochondrial copy number compromise normal mitochondrial function and promote cellular migration, thus contributing to tumor progression [53]. Accordingly, the fragmentation-inducing effect of DEA (Figure 3) may contribute to its cytotoxic properties and ability to limit in vivo metastasis, which we observed previously in cultured bladder and cervix carcinoma and melanoma cells [15,16,17]. Increased mitochondrial fragmentation results from increased fission or decreased fusion. Considering that fusion requires intact ΔΨ_m_ [29], and DEA-impaired ΔΨ_m_ (Figure 2), it seems likely that DEA caused mitochondrial fragmentation by impeding fusion. Opa1 is responsible for the fusion of the IMM, and maintains its cristae morphology [29]. Short and long isoforms of the protein are in balance, and normally they are associated with the IMM. However, their balance is disrupted, and the protein is released to the cytosol when stimuli disturb the ΔΨ_m_ and/or permeabilize the OMM [30]. DEA at a concentration of 10 µM evoked Opa1 release to the cytosol (Figure 6), indicating that its mitochondrial fragmentation-inducing effect was indeed caused by impeding fusion rather than by promoting fission.

Cancer cells change their metabolism in characteristic ways to adapt to the predominantly hypoxic conditions of their environment [54]. For energy production, many cancer types prefer glycolysis over mitochondrial oxidative phosphorylation, even in the presence of sufficient oxygen. The importance of this switch in ATP production, known as the Warburg effect [55], is validated by the diagnostic value of ^18^F-deoxyglucose positron emission tomography in identifying tumors based on their increased glucose uptake [56,57]. However, the most malignant cancer types, including cancer stem cells, metastatic tumor cells, and therapy-resistant tumor cells, have elevated levels of mitochondrial ATP synthesis [58,59]. Moreover, oxidative phosphorylation is essential for the survival, proliferation, and metastasis of these cells and forms the basis of their resistance to chemotherapy and radiotherapy [60,61]. Accordingly, oxidative phosphorylation is considered as an emerging therapeutic target, especially for the most malignant cancer types [62]. Based on its effects on energy metabolism in B16F10 melanoma cells, DEA exhibits promise as a candidate cancer therapy. At a concentration of 10 µM, DEA diminished maximal respiration, ATP production, and coupling efficiency (Figure 4). These findings are in line with our previous results obtained in isolated liver and heart mitochondria [18]. Additionally, DEA decreased fermentative glycolysis and non-mitochondrial oxygen consumption in a concentration-dependent manner (Figure 4). This latter effect might contribute to the drug’s cytotoxic and anti-metastatic properties. Impeding both the mitochondrial and glycolytic ATP producing pathways (Figure 4) seems an attractive mechanism for DEA’s cytotoxic property. However, this remains to be proven in future studies aimed at identifying the targets of DEA both in the respiratory chain and the glycolytic pathway.

Adverse environmental conditions such as hypoxia, low nutrient availability, and growth factor withdrawal can induce mitochondria-associated cell death, most often in the form of mPT or OMM permeabilization. When opened, the mPT pore forms a nonspecific channel, allowing the unrestricted passage of water and solutes of up to 1.5 kDa in size across the IMM [26]. The physiological role of transient channel opening remains unknown; however, prolonged opening results in the complete loss of ΔΨ_m_, the termination of ATP synthesis, mitochondrial swelling, and the rupture of the IMM due to water influx driven by the high osmolarity of the mitochondrial matrix [26,63]. Outflowing Ca^2+^ triggers mPT in the neighboring mitochondria, resulting in eventual necrotic cell death [26]. However, cancer cells are often relatively resistant to mPT induction [64]. In complete agreement with our previous results obtained in isolated liver and heart mitochondria [18], 10 µM of DEA induced a CsA-independent mPT (Figure 5). This likely contributes to DEA’s cytotoxicity in cultured bladder carcinoma, cervical carcinoma, and melanoma cells [15,16,17]. Furthermore, DEA definitely compromised IMM integrity, as calcein fluorescence was not quenched in the absence of DEA or Ca^2+^ stimulation (data not shown). In addition, Ca^2+^ evoked a CsA-dependent mPT that was more pronounced than the mPT induced by 10 µM of DEA (Figure 5). The identity and composition of the mPT pore remains controversial [63], and CsA-independent mPT induction has been reported by others [65]. Therefore, we will refer to this effect of DEA as “mPT-inducing” until a clearer definition of mPT can be established.

The other major form of mitochondria-associated cell death is OMM permeabilization-mediated apoptosis. However, resistance to apoptosis is a hallmark of cancer that is achieved by disrupting the balance between pro- and anti-apoptotic Bcl2 proteins [66]. Resistance arises due to a multitude of mechanisms, including the downregulation of pro-apoptotic, up-regulation of anti-apoptotic Bcl-2 genes [67], and upregulation of protein inhibitors of apoptosis [68]. Although only preliminary data are available regarding how DEA interferes with these mechanisms, some trends have emerged. Firstly, even a relatively short DEA treatment (6 h) elevates the steady-state level of the pro-apoptotic Bcl-2 family member Bad (Figure 6). Future studies should investigate whether Bad levels rise due to the upregulation of expression or the inhibition of degradation. Second, the Akt, which is constitutively active in melanomas [69], was inactivated by 10 µM of DEA (Figure 6). In this case, the duration was more than adequate for the observed effect, as the covalent regulation of a kinase activity could be achieved within minutes. The observed decrease in Bad phosphorylation and increase in Cyt c and Aif release (Figure 6) could be explained by the previous two findings.

However, the data in Figure 6 also indicate that several elements of the signaling network underlying DEA’s OMM-permeabilizing effect were not examined in this study. Although it is difficult to compare blots using different primary antibodies, it seems unlikely that the observed Akt inhibition can fully account for the reduction in Bad phosphorylation (Figure 6). Additionally, the extent of Cyt c release seems disproportional with the other observed changes (Figure 6). Future studies should seek to identify missing mechanisms. On the other hand, all the aforementioned data are in line with our previous findings on DEA [15,16,17,18], and they account for the time- and concentration-dependent cytostatic effect of DEA treatment (Figure 7). Based on the data presented in this study, it is difficult to propose a mechanism that can explain the said multiple effects on several mitochondrial processes. Considering the central role of Ca^2+^ in regulating apoptosis and thereby the survival and therapy resistance of cancer cells [70], it is a limitation of this study that it cannot account for DEA’s effect on intracellular calcium.

On the other hand, we presented experimental evidence that DEA decreases ΔΨ; induces mitochondrial fragmentation; decreases maximal respiration, ATP production, coupling efficiency, glycolysis, and non-mitochondrial oxygen consumption; and induces CsA-independent mPT and OMM permeabilization. All of these effects may account for the rapid (3–12 h) cytotoxicity of the drug. Furthermore, these results are in agreement with our previous findings obtained in isolated liver and heart mitochondria, and may account for DEA’s long-term (24–72 h) cytotoxicity and ability to suppress in vivo metastasis by bladder carcinoma, cervical carcinoma, and melanoma cell lines [15,16,17,18].

## 4. Materials and Methods

### 4.1. Materials

Protease inhibitor cocktail and all chemicals for cell culture were purchased from Sigma-Aldrich Kft (Budapest, Hungary). DEA was a gift from Professor Andras Varro (Department of Pharmacology and Pharmacotherapy, University of Szeged, Szeged, Hungary). The following primary antibodies were used: anti-Bad, anti–phospho-Bad (Ser136), anti- Akt, anti-phospho-Akt (Ser473), anti-AIF, anti-histonH1, anti-cytochrome C, anti–Opa1 (1:500 dilution), anti-GAPDH (1:2000, clone 6C5), and anti-actin (1:2000). All the antibodies were purchased from Cell Signaling Technology (Beverly, MA, USA), but GAPDH that was from EMD Millipore Bioscience (Darmstadt, Germany).

### 4.2. Cell Culture

B16F10 mouse metastatic melanoma cells were obtained from the American Type Culture Collection (LGC Standards, Wesel, Germany). B16F10 cells were split twice a week for 4 months and maintained as monolayer adherent cultures under standard conditions (5% CO2, 37 °C) in RPMI 1640 media supplemented with 10% fetal calf serum (FCS) and 1% penicillin–streptomycin mixture (Life Technologies, Darmstadt, Germany).

### 4.3. Cell Viability Assay

B16F10 cells were seeded in 96-well plates at a starting density of 5 × 10^3^ cells/well in quintuplicate (five replicate wells per sample) overnight. The cells were treated with 0, 5, or 10 µM of DEA for 3–12 h. After treatment, the cells were washed in phosphate-buffered saline (PBS) and fixed in 100 µL of cold 10% trichloroacetic acid solution. The plates were incubated for 30 min at 4 °C, washed five times with distilled water, and dried overnight at room temperature. The cellular protein content in the wells was determined by the SRB assay. Briefly, 70 µL of 0.4% SRB (Sigma-Aldrich Co., Budapest, Hungary) prepared in 1% acetic acid were added to each well and incubated for 30 min at room temperature. The SRB reagent was discarded, and the plates were washed five times with 1% acetic acid and dried at room temperature for a few hours. Then, 200 µL of a 10 mM Tris base was added to each well, and the samples were incubated at room temperature on a plate shaker for 30 min to solubilize the bound SRB. Absorbance was measured at 560 and 600 nm in parallel on a plate reader. OD_600_ was subtracted as a background from the OD_560_. These experiments were repeated five times.

### 4.4. Bioenergetic Analysis

To determine the balance of oxidative vs. fermentative energy production in B16F10 cells, OCR and ECAR were measured using a Seahorse XFp Analyzer (Agilent, Santa Clara, CA, USA). Cells were seeded into XFp cell culture 8-well miniplates at a starting density of 4 × 10^4^ cell/well in duplicate and cultured under standard conditions overnight. Then, the cells were treated with 0, 5, or 10 µM of DEA for 3 h. Prior to the measurement, the medium was replaced with Seahorse XF Assay Media (Agilent, Santa Clara, CA, USA) pH 7.4 supplemented with 10 mM of glucose, 2 mM of L-glutamine, and 1 mM of pyruvate. Mitochondrial stress test was performed using the following inhibitors at the indicated final concentrations: 1 µM of oligomycin, 1 µM of FCCP, and 1 µM of rotenone–antimycin A. In each experiment, two wells without cells were running to assess the non-cellular oxygen consumption, which was subtracted from the corresponding OCR value. The OCR and ECAR data were normalized to the mg protein content using the Micro BCA Protein Assay kit (Thermo Fisher Scientific, Waltham, MA, USA) for the measurement of protein concentrations. No other data correction was applied. These experiments were repeated three times.

### 4.5. ΔΨ_m_ Assay

Changes in ΔΨ_m_ were assayed using the mitochondrial fluorescent dye JC-1. B16F10 cells were seeded at a starting density of 2.5 × 10^4^ cells/well in 6-well plates containing coverslips and cultured at least overnight before the experiment. After subjecting the cells to different concentrations of DEA for 3 h, the coverslips were rinsed twice in PBS and placed upside down on top of a small chamber formed by a microscope slide filled with PBS supplemented with 10% FCS and 1 μg/mL of JC-1 dye (Molecular Probes, Eugene, OR, USA). The cells were imaged with a Nikon Eclipse Ti-U fluorescent microscope (Auro-Science Consulting Ltd., Budapest, Hungary) equipped with a Spot RT3 camera, using a 20× objective lens with epifluorescence illumination. After the cells were loaded with dye for 15 min, the same microscopic field was imaged with a 490 nm bandpass excitation and > 590 nm (red) and < 546 nm (green) emission filters. Under these conditions, we did not observe considerable bleed-through between the red and green channels. The quantification of JC-1 fluorescence intensities in each sample was performed in 15 randomly chosen microscopic fields containing 20–30 cells using the MetaXpress image analyzer software (Molecular Devices LLC., San Jose, CA, USA). These experiments were repeated three times.

### 4.6. Analysis of Mitochondrial Network Dynamics

For confocal imaging, B16F10 cells were seeded onto 25 mm round glass coverslip medium at a starting density of 2.5 × 10^4^ cells/coverslip, and cultured in antibiotic-free culture for 24 h. The transient transfection of B16F10 cells with mtRFP was performed using TransFectin Lipid Reagent (Bio-Rad, Hercules, CA, USA). On the following day, the cells were treated for 3 h, as indicated in the text; washed twice in PBS; and fixed in 4% formalin. Fluorescence was visualized on an Olympus FluoView 1000 (Olympus, Hamburg, Germany) confocal laser scanning microscope. For excitation, a multiline argon-ion laser at 488 nm and a green helium-neon laser at 543 nm were used (10 µs/pixel) in the photon-counting and sequential mode. The field of interest was scanned in XYZ mode, scanning the total thickness of the cells with a 1.5 µm layer distance and taking 1024 × 1024 pixel images of each layer. The quantitative determination of mitochondrial fragmentation was performed as described [25]; we considered mitochondria shorter than 2 μm to be fragmented and those longer than 5 μm as filamentous. These experiments were repeated three times.

### 4.7. Subcellular Fractionation

Three semi-confluent 10 cm plates of B16F10 cells were harvested, washed twice with PBS, and resuspended in 1 mL of fractionation buffer (250 mM of sucrose, 20 mM of 2-[4-(2-hydroxyethyl)piperazin-1-yl]ethanesulfonic acid (HEPES), pH 7.4, 10 mM of KCl, 1.5 mM of MgCl_2_, 1 mM of ethylenediamine-tetraacetic acid (EDTA), 1 mM of ethyleneglycol-tetraacetic acid (EGTA), 1 mM of dithiothreitol (DTT), and proteinase inhibitor cocktail (Sigma, #P2714)). The cell lysate was manually homogenized in a Teflon–glass homogenizer, chilled on ice, and centrifuged for 7 min at 720× *g* at 4 °C. The nuclear pellet was resuspended in 700 μL of fractionation buffer, re-homogenized, and centrifuged for 10 min at 600× *g*. This procedure was repeated one more time, and the pellet was resuspended in lysis buffer (10% glycerol, 25 mM of NaCl, 50 mM of NaF, 10 mM of Na-pyrophosphate, 2 nM of EGTA, 2 nM of DTT, 20 nM of p-nitrophenyl-phosphate, 25 mM of Tris-HCl, pH 7.4, 50 nM of beta-glycerophosphate, and 0.1% Triton X-100) to yield the nuclear fraction. The supernatant of the 7 min, 720× g centrifugation step was subjected to centrifugation at 10,000× *g* for 5 min at 4 °C. The supernatant and pellet of this step yielded the cytoplasmic and mitochondrial fractions, respectively; the latter was not used in this study.

### 4.8. Immunoblot Analysis

B16F10 cells were seeded in 10 cm plates at 10^6^ cells/plate and treated as described above for the cell viability assay. The cells were harvested at intervals in chilled lysis buffer containing 0.5 mM of sodium-metavanadate, 1 mM of EDTA, and protease inhibitor cocktail (1:200). The cell lysates were boiled and subjected to 10% sodium dodecyl sulfate polyacrylamide gel electrophoresis before being transferred to nitrocellulose membranes. The membranes were blocked in 5% low-fat milk for 1.5 h at room temperature, and then exposed to primary antibodies at 4 °C overnight in blocking solution. Appropriate horseradish peroxidase–conjugated secondary antibodies were used at a dilution of 1:5000. Signals were visualized by using enhanced chemiluminescence and captured on X-ray film. The films were scanned, and the pixel densities of the bands were determined using the NIH ImageJ software. Alternatively, chemiluminescence was measured on an Azure 300 (Azure Biosystems) imaging system that digitized the bands’ chemiluminescence intensities using its inbuilt software. For stripping and reprobing, the membranes were washed in stripping buffer (0.1 M glycine, 5 M MgCl_2_, pH 2.8) for 1 h at room temperature. After washing and blocking, the membranes were incubated with primary antibodies against non-phosphorylated or loading control proteins. These experiments were repeated three times.

### 4.9. Determination of Cellular ROS Formation

B16F10 cells were seeded as described above for the viability assay and treated with 5 or 10 µM of DEA or 5 µM of taxol (positive control [21]). The cellular ROS levels were determined based on the formation of N-acetyl-8-dodecyl-resorufin and 6-carboxy-2’,7’-dichlorofluorescein from their nonfluorescent reduced counterparts, N-acetyl-8-dodecyl-3,7-dihydroxyphenoxazine (1.3 mg/L final concentration) and 6-carboxy-2’,7’-dichlorodihydrofluorescein diacetate (2 mg/L final concentration), as described previously [21,71]. The former assessed the cellular ROS formation in the lipid phase, and the latter in the aqueous phase. Fluorescence was measured by a plate-reader fluorimeter (PerkinElmer, Hungary) at excitation wavelengths of 578 and 495 nm and emission wavelengths of 597 and 522 nm for N-acetyl-8-dodecyl-resorufin and 6-carboxy-2’,7’-dichlorofluorescein, respectively. The mitochondrial superoxide formation was assessed by MitoSOX^TM^ Red (Thermo Fisher Scientific, Waltham, MA, USA). Cells were incubated for 30 min in 5 µM of MitoSOX^TM^ Red, then were rinsed and treated with 5 or 10 µM of DEA or 5 µM of taxol for 3 h. Fluorescence was recorded by the plate-reader fluorimeter at excitation wavelengths of 495 nm and emission wavelengths of 590 nm. Under these conditions, the fluorescence intensity is proportional to the mitochondrial superoxide level [20]. The ROS levels were calculated from the slopes of the registration curves. These experiments were repeated five times.

### 4.10. Measurement of mPT in Intact B16F10 Cells

The B16F10 cells were seeded as described above for the viability assay and cultured in antibiotic-free medium overnight. Cells were washed with Ca^2+^- and Mg^2+^-free Hank’s balanced salt solution (HBSS), and then treated with 250 nM of A23187, 90 µM of CoCl_2_, 1 g/L of glucose, 5 or 10 µM of DEA or 1.5 mM of CaCl_2_ (positive control [27]), and/or 2.5 µM of CsA in HBSS for 3 h. For the detection of mPT, acetoxymethylcalcein was added to the medium at a final concentration of 1 µM. Fluorescence in wells resulting from unquenched de-esterified calcein in the mitochondria [27] (cytoplasmic calcein is quenched by Co^2+^) was monitored using the ImageXpress Micro4 automated high-content imaging system (Molecular Devices LLC., San Jose, CA, USA) using a 4× objective and epifluorescence illumination. The quantification of calcein fluorescent intensities was performed using the MetaXpress software. These experiments were repeated three times.

### 4.11. Data Analysis

All the data are expressed as means ± standard deviation (SD). The concentration-dependent effects of DEA in each experiment were tested with ANOVA using the post hoc Dunnett test. Differences were considered significant at *p* < 0.05. Statistical analyses were performed using IBM SPSS Statistics v20.0.

## Figures and Tables

**Figure 1 ijms-21-07346-f001:**
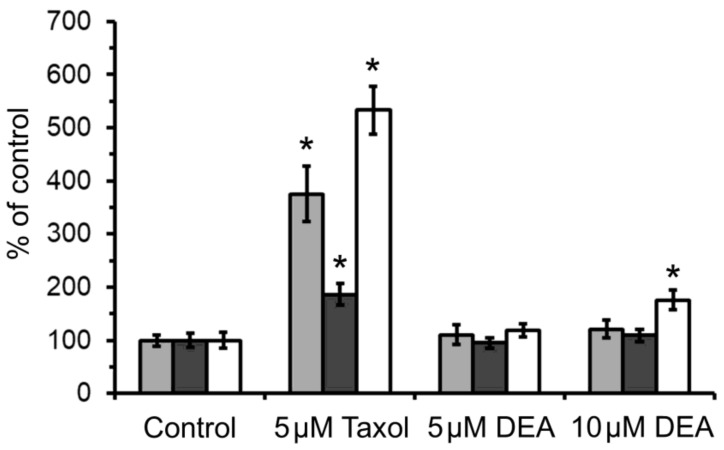
Effect of DEA on the cellular ROS formation in the B16F10 melanoma line. Cells were treated with 5 or 10 µM of DEA or 5 µM of taxol (positive control). ROS formation in the lipid (light bars) and aqueous (dark bars) phase and mitochondrial superoxide production (empty bars) were calculated based on the rate of fluorescence intensity change vs. time; fluorescent dyes were generated by ROS-mediated oxidation from their respective non-fluorescent counterparts. The results are expressed as the % of ROS formation in the absence of agents (means ± SEM of five independent experiments). * indicates a significant difference relative to the control (*p* < 0.05).

**Figure 2 ijms-21-07346-f002:**
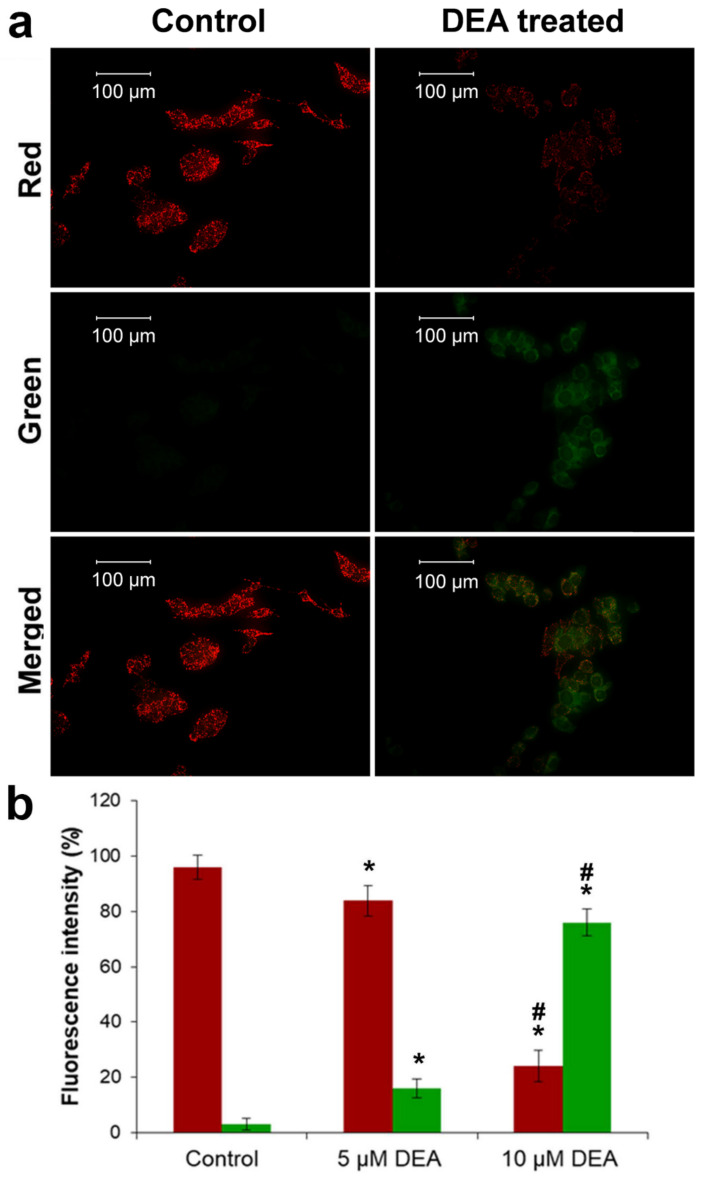
Effect of DEA on ΔΨ_m_ in B16F10 melanoma cells. Cells were treated with 5 or 10 µM of DEA for 3 h. ΔΨ_m_ was assessed using the membrane potential-dependent fluorescent dye, JC-1. Red and green fluorescence indicates normal and depolarized ΔΨ_m_, respectively. (**a**) Representative fluorescence images in the red, green, and merged channels of cells treated with 10 µM of DEA. (**b**) Quantitative assessment of ΔΨ_m_, expressed as the % of fluorescence intensity (means ± SEM of three independent experiments). Quantitative comparisons are true within the same color only. Red and green bars denote red and green fluorescence, respectively. * and # indicate a significant difference relative to the control and 5 µM of DEA, respectively (*p* < 0.05).

**Figure 3 ijms-21-07346-f003:**
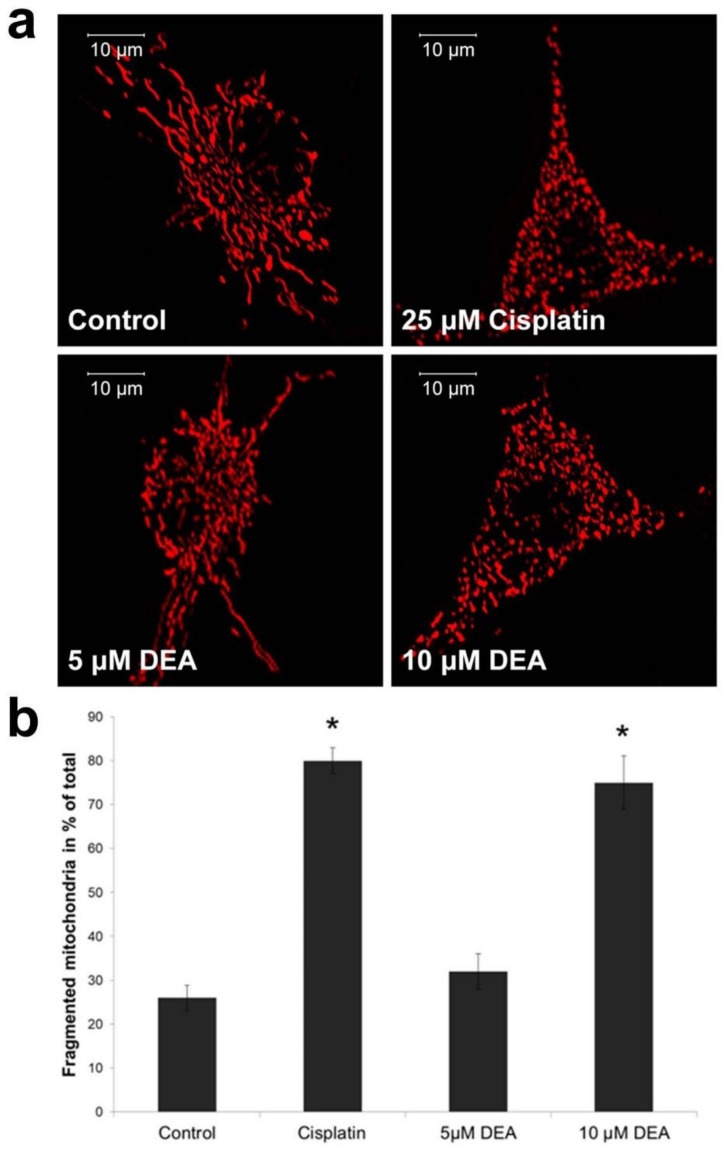
Effect of DEA on mitochondrial fragmentation. mtRFP-transfected cells were treated with 5 or 10 µM of DEA, or 25 µM of cisplatin (positive control [24]), for 3 h. Mitochondrial fragmentation was determined based on confocal fluorescence images, as described in [25]. (**a**) Representative fluorescence images for all the treatment groups. (**b**) Quantitative assessment of mitochondrial fragmentation expressed as % (means ± SEM of three independent experiments). * indicates a significant difference relative to the control (*p* < 0.05).

**Figure 4 ijms-21-07346-f004:**
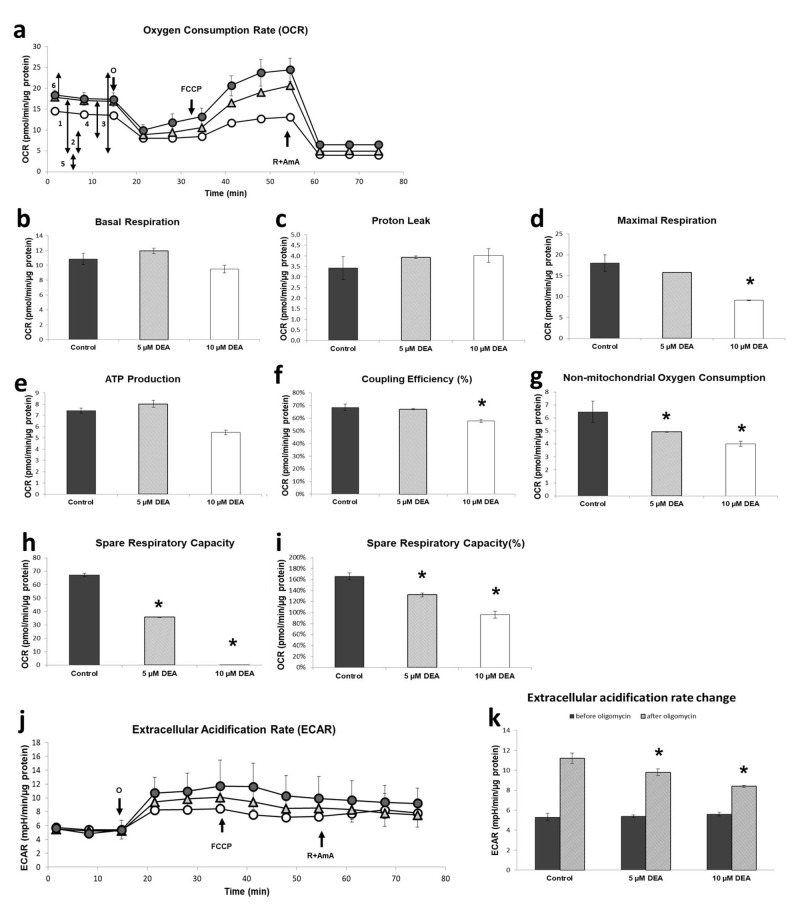
Effect of DEA on the energy metabolism of B16F10 melanoma cells. Cells were treated with 5 or 10 µM of DEA for 3 h, and then the OCR and ECAR were monitored for 75 min. The F_o_F_1_ ATP synthase inhibitor oligomycin (o), the uncoupler FCCP, and the respiratory inhibitors rotenone and antimycin A (R+AmA) were added at the bold arrows. (**a**) OCR recordings for untreated (filled circles), 5 µM of DEA-treated (triangles), and 10 µM of DEA-treated (open circles) cells (means ± SD of three independent experiments running in two replicates in each experiment. The double-headed arrows with numbers next to them indicate (1) basal respiration, (2) proton leak, (3) maximal respiration, (4) ATP production, (5) non-mitochondrial oxygen consumption, and (6) spare respiratory capacity. (**b–i**) Parameters derived from (**a**); for explanation, see the text and (**a**). Appropriate parts of the recordings were averaged and presented as the means ± SEM of three independent experiments running in two replicates in each experiment. * indicates a significant difference relative to the control (*p* < 0.05). (**b**) Basal respiration. (**c**) Proton leak. (**d**) Maximal respiration. (**e**) Mitochondrial ATP production. (**f**) Coupling efficiency; for explanation, see the text. (**g**) Non-mitochondrial oxygen consumption. (**h,i**) Spare respiratory capacity, presented as the difference (**h**) and ratio (**i**) of maximal and basal respiration. (**j**) ECAR recordings for untreated (filled circles), 5 µM of DEA-treated (triangles), and 10 µM of DEA-treated (open circles) cells (means ± SD of three independent experiments running in two replicates in each experiment). All the labeling is the same as for (**a**). (**k**) Extracellular acidification rate before (dark bars) and after (light bars) the administration of oligomycin. OCR and ECAR data were normalized to mg of protein content.

**Figure 5 ijms-21-07346-f005:**
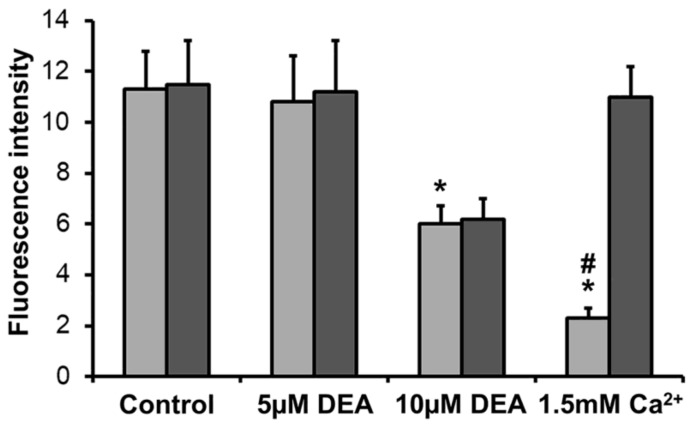
Effect of DEA on mPT in intact B16F10 melanoma cells. Cells were treated with 5 or 10 µM of DEA or 1.5 mM of Ca^2+^ (positive control [27]) in the presence (dark bars) or absence (light bars) of the mPT inhibitor CsA for 3 h. mPT was assessed by monitoring the Co^2+^-mediated quenching of mitochondrial calcein fluorescence [27]. The results are expressed as fluorescence intensity (the means ± SEM of three independent experiments). * and # indicate a significant difference relative to control and CsA-treated cells of the same treatment group, respectively (*p* < 0.05).

**Figure 6 ijms-21-07346-f006:**
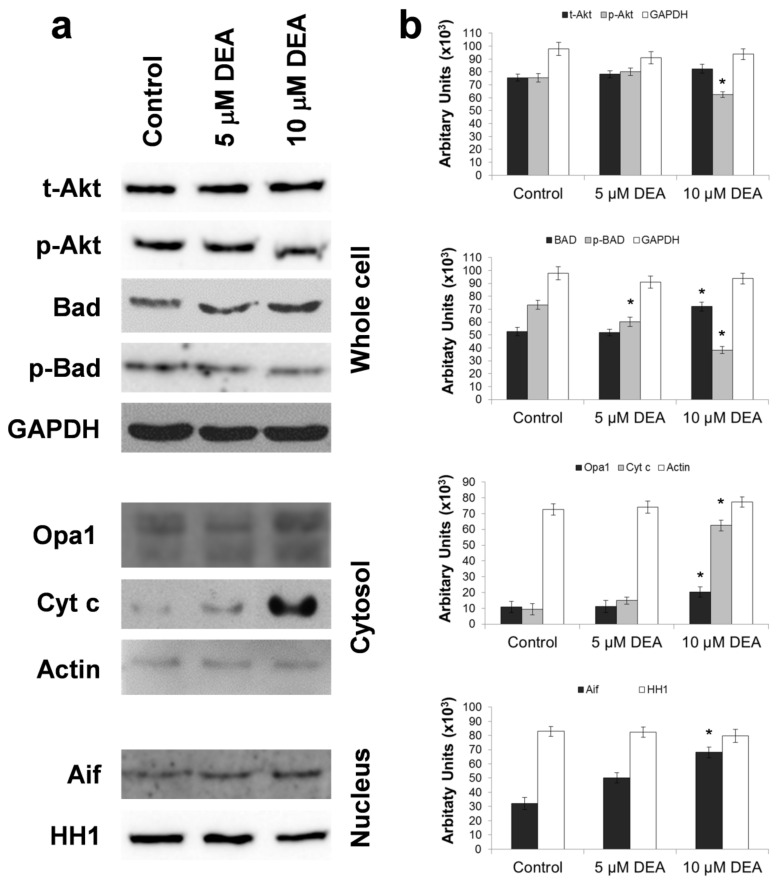
Effect of DEA on the OMM permeabilization in B16F10 melanoma cells. Cells were treated with 5 or 10 µM of DEA for 6 h, harvested, and homogenized (Whole cell). Alternatively, nuclear and cytosolic fractions were prepared from the harvested cells. Steady-state levels and phosphorylation states of Akt and Bad were assessed in whole-cell homogenate. The steady-state levels of Cyt c and Opa1 were determined in the cytosolic, whereas that of Aif was measured in the nuclear fraction by immunoblotting. In the whole-cell homogenate, the cytosolic fraction and the nuclear fraction, we used glyceraldehyde-3-phosphate dehydrogenase (GAPDH), actin and histone H1 (HH1), respectively, as loading controls. The proteins were visualized by enhanced chemiluminescence. (**a**) Representative blots. (**b**) Quantitative assessment of proteins in the subcellular fractions. Results are expressed as pixel density or chemiluminescence intensity of the bands, both expressed in arbitrary units (means ± SEM of three independent experiments). * indicates a significant difference relative to the control (*p* < 0.05).

**Figure 7 ijms-21-07346-f007:**
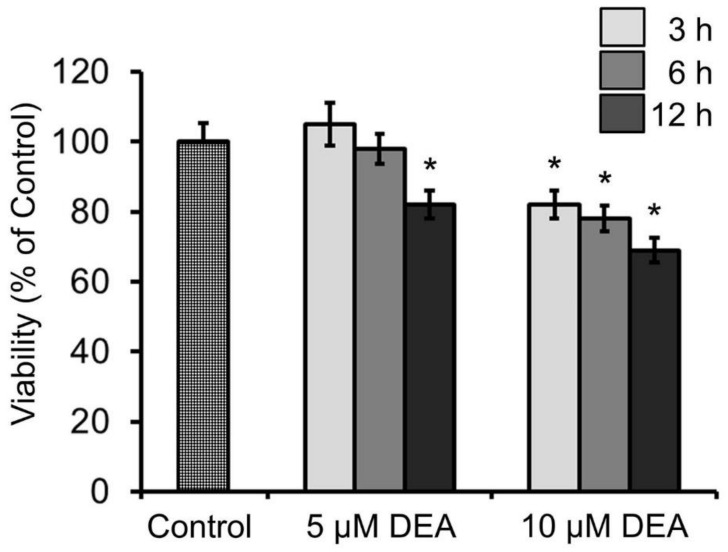
Effect of DEA on the viability of B16F10 melanoma cells. Cells were treated with 5 or 10 µM of DEA for 3–12 h, and then the viability was determined using the SRB assay. The results are expressed as the % viability of the control (means ± SEM of five independent experiments). * indicates a significant difference relative to the control (*p* < 0.05).

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
