# Peer review of "Involvement of Mitochondrial Mechanisms in the Cytostatic Effect of Desethylamiodarone in B16F10 Melanoma Cells"

_ijms, 2020, doi:10.3390/ijms21197346_

Round 1

Reviewer 1 Report

This is an interesting and thought provoking paper that turns a potential metabolite-induced side effect into a potential treatment for cancer. The targeting of mitochondria in cancer is a burgeoning research focus, and is thus very timely, especially by using a compound derived from such a well-known drug. This concept alone is worth publishing. With regards the manuscript, it is mostly well written, but there a number of problems with the English in places. There are also a couple of mechanistic points that could also be discussed that would help strengthen it. Methodologically, the major weakness is not having compared a non-cancerous cell line, and thus, being able to address dose-differences between healthy and cancerous cells. However, this point could be addressed by a slightly more detailed discussion. Specific points to address:

  • Get a really good English editor to through it as some of the spelling and phraseology is a bit odd, e.g., “As we found” is not really applicable. Punctuation wrong in places, as is the capitalisation, and the spacing (e.g., Cyt C, not CytC, and standardisation of spacing when using units)
  • Line 39, what concentrations does this metabolite reach in tissues in vivo?
  • Figure 2, images a bit faint for publication (but are perfect in figure 3)
  • Line 232. A lot of ROS is produced by signalling pathways not of mitochondrial origin, and mitochondria do actually very tightly control ROS production, even in cancer. So perhaps should revisit this section in light of newer data. Also need to be everso careful about making sweeping statements about Nrf2 – there are many redox control systems. Plus, need to explain how inhibiting mitochondrial function can inhibit cancer, even if it doesn’t induce ROS. What is your underlying theory here?
  • Line 255. Mitochondrial dynamics, cause and effect. Do mitochondria fragment because they are damaged, or to prevent damage or alter signalling, and how do they react in hypoxia and why? The key point here is that mitochondrial function in modulated during cancer, and is extremely important. This is a complex subject and perhaps needs a little more discussion. Also have to remember that mitochondrial location in the cell plays a role, and is highly variable between different cancers, and even within the life cycle of the cell. Are we actually seeing increased mitophagy, or simply fragmentation (so may need to consider how to answer this). The decreased respiratory capacity suggests a massive decrease in functional mitochondria, which seems to go hand in hand with reduced glycolysis, how could this be? The mitochondria don’t seem to be swollen at this time point, but do they at later time points?
  • Line 270. What about the role of HK and VDAC? It is becoming clear that the Warburg effect is only part of the life cycle of the cancer cell, and to metastasize, it ups respiration again. Again, need to discuss role of concentration here and what would happen with a non-cancerous cell. May be worth referencing the Bennett 2020 paper on the ATPome (provides some very interesting insight into how respiration and glycolysis are balanced).
  • Line 288. The main point missing here is that cancer cells evolve and redirect calcium away from mitochondria, which neutralises, to some extent, apoptosis. This is another hot button in this field, so should at least be mentioned, and may well help explain how this drug works. E.g., Danese, 2017.
  • Line 327. How pure is the DEA?
  • Line 406. What does this sentence actually mean?

Line 450. Why is the conclusions bit here? May be worth suggesting a mode of action, possibly with a cartoon graphic.

Author Response

We would like to thank the reviewer his/her thorough work on the manuscript. We believe that his/her contribution has improved it considerably.

General consideration

We agree with the reviewer in that comparison of DEA’s effect on melanoma vs. normal cells indeed would be informative. However, the aim of this study was determining whether mitochondrial mechanisms were involved in the cytostatic effect of DEA. Therefore, we feel that involving normal cells in the experimental setups exceeds scope of the present study. Additionally, performing such experiments would need much more time than the 10 days we were allotted for revision.

We tried to comply with all other requests regarding improving the quality of the manuscript’s English, and revising the discussion.

Major points

Bioedit Ltd. https://www.bioedit.com/ performed editing of the manuscript. We ignored some of their comments: CP2 - catabolism is not equal to anabolism (synthesis of essential intermediates and macromolecule precursors for cell growth and proliferation); CP6 – we think the information is covered by “mPT was assessed by monitoring Co2+-mediated quenching of mitochondrial calcein fluorescence”; CP7 – A23187 when mentioned first was described as a calcium ionophore facilitating Co2+ uptake; CP12 – we do confirm low-fat milk; CP13 – we do not agree. We accepted all other comments and corrections.  

Ad line 39: 100-1000 times higher than the corresponding plasma concentration according to reference 11. In cell culture, incubation in 10 µM DEA resulted in mM intracellular concentration of the drug according to reference 13.

Ad Figure 2: We replaced the fluorescence images to ones of better quality as requested by the reviewer.

Ad line 232: We managed to detect DEA-induced mitochondrial superoxide production by using MitoSOXTM Red, although the extent of superoxide production was less than 20% of the one induced by Taxol, the positive control. We added the new data to Figure 1, and modified the Results, Discussion and Materials and Methods sections accordingly.

Ad line 255: To address the question whether mitochondrial fragmentation-inducing effect of DEA was achieved by impeding fusion or promoting fission, we assessed its effect on Opa1 release. The new data added to Figure 6 supported the latter possibility. We modified the Results, Discussion and Materials and Methods sections accordingly.
In this study, we focused on early events in DEA’s cytotoxic effect, but obviously these events have to be revisited at later time points.

Ad line 270: Cancer cells usually regulate mitochondrial and glycolytic ATP production in a reciprocal way according to their metabolic need. In contrast, DEA treatment impeded both ATP producing pathways (Figure 4). However, one should consider that DEA treated cancer cell eventually die likely for the very reason of unregulated energy metabolism. Clearly, the targets of DEA both in the respiratory chain and the glycolytic pathway should be identified to support the said hypothesis.
HK and VDAC were assumed to be part of the mPT pore about a decade ago, and they may have role in mediating DEA’s effect. Certainly, further studies are needed to establish such a role for them.

Ad line 288: We fully agree with the reviewer in the importance of calcium in regulating apoptosis thereby survival and therapy resistance of cancer cells. It is a major limitation of this study, and DEA’s effect on intracellular calcium should be determined in further studies. We indicated this fact in the Discussion, and added the suggested reference.

Ad line 327: The DEA we used is >99% pure according to mass spectrometry.

Ad line 406: Hopefully, the sentence has been clarified.

Ad line 450: We feel that this study raised a number of questions that require some additional experiments before a conclusive mechanism of action can be proposed. Accordingly, we moved the paragraph to the Discussion section and refrained to draw conclusions.

Reviewer 2 Report

The DEA, a major metabolite of Amiodarone, is already reported inducing apoptosis on cancer cells ( Bognar et al., PLoS One. 2017; 12(12. Bognar et al. J Physiol Pharmacol. 2018 Oct;96(10):1004-1011; ). The manuscript by Ramadan et al. confirms some effects of DEA, on mitochondria (see Bolt et al.  J Pharmacol Exp Ther . 2001 Sep;298(3):1280-9; . Nicolescu et al. Toxicol Appl Pharmacol. 2008 Mar 15;227(3):370-9; J T Di Matola  et al.  Clin Endocrinol Metab . 2000 Nov;85(11):4323-30) and speculates that these effects may account for its cytotoxic and in vivo metastasis-limiting properties. For this purpose the effect of DEA was analyzed in  B16F10 melanoma cells.

General consideration

The cancer cells exhibit hybrid metabolisms and  high level of metabolic plasticity.  The different tumor cells are characterized by a great metabolic heterogeneity in which glycolysis and the mitochondrial respiratory chain can have a different weight in the energy supply, for this reason it would be useful to characterize the bioenergetics profile of B16F10 melanoma cells (compare to control cells). The effect of hypothetic drug molecules could be different depending on basal bioenergetics profile of cancer cells.

Please see below my specific comments:

 Major points

  • 1 The measurement of ROS increase with N-acetyl-8-dodecyl-3,7-dihydroxyphenoxazine and 6-carboxy-2',7'-dichlorodihydrofluorescein diacetate, not revealed ROS increase after DEA treatment. Mitochondrial dysfunction is generally associates with mitochondrial ROS production, indeed I suggest to use specific mitochondrial probe to verify this aspect.
  • In the fig 2 the effect of DEA on mitochondrial membrane potential is presented, the resolution of the images should be improved, the objective used for magnification should be indicated. In addition an image in which the unlabelled cells are show must be added to normalized the JC1 signal.
  • In the bioenergetics analysis the data reported in histograms must be normalize for cells number or mg total protein. In addition the authors should described if the values ​​of OCR and ECAR were obtained after correction for residual activity in the presence of the mitochondrial respiratory chain inhibitors rotenone plus antymicin A and in the presence of the glycolysis inhibitor 2-deoxyglucose for ECAR.
  • the reported data should be enriched by the functional and / or structural analysis of the mitochondrial respiratory chain complexes (complex I, II + III, IV and V)
  • The effects of DEA on mitochondrial fragmentation should be accompanied by the analysis of the proteins involved in this process (DRP1, P-DRP1 etc…).

Author Response

We would like to thank the reviewer his/her thorough work on the manuscript. We believe that his/her contribution has improved it considerably.

General consideration

We fully agree with the reviewer in the importance of metabolic heterogeneity of cancer cells. Accordingly, comparison of DEA’s effect on melanoma vs. normal cells indeed would be informative. However, the aim of this study was determining whether mitochondrial mechanisms were involved in the cytostatic effect of DEA. Therefore, we feel that involving normal cells in the experimental setups exceeds scope of the present study. Additionally, performing such experiments would need much more time than the 10 days we were allotted for revision.

Major points

 1/ We are indebted to the reviewer for suggesting the use of mitochondria specific redox dye. Indeed we managed to detect mitochondrial superoxide production by using MitoSOXTM Red, although the extent of superoxide production was less than 20% of the one induced by Taxol, the positive control. We added the new data to Figure 1, and modified the Results, Discussion and Materials and Methods sections accordingly.

2/ We replaced the fluorescence images to ones of better quality as requested by the reviewer. The images’ resolution was limited by the objective’s power (20x), which was selected to accommodate 10-20 cells in the objective’s field of vision. We felt superfluous to indicate the objective’s magnification in the figure legend, since a scale was presented in each image. We agree with the reviewer in incomparability of green and red fluorescence. Strictly speaking, presenting green and red fluorescence as % of total fluorescence can be misleading (since 1 unit of green fluorescence is not equal to 1 unit of red fluorescence). However, we just followed the customary way of presenting JC-1 data. We also added a sentence to the figure legend to warn the readers of comparing fluorescence intensity data just within the same colours.

3/ We recalculated OCR and ECAR values after normalizing the data to mg protein content as requested by the reviewer. We presented average OCR and ECAR registration curves (Figure 4a and j) generated by the Seahorse software. In each experiment, two wells without cells were running to assess non-cellular oxygen consumption, which was automatically subtracted from the corresponding OCR value by the software. We also indicated that we did not apply any additional data correction.

4/ We think that studying DEA’s effect on functional and / or structural aspects of the mitochondrial respiratory chain complexes could be interesting, but we feel that it exceeds scope of the present study. Additionally, performing such studies would need much more time than the 10 days we were allotted for revision.

5/ Although mitochondrial fission is caused by DRP1, DEA’s effect on ΔΨ and mPT suggested that the observed fragmentation was due to diminished fusion rather than activated fission. Accordingly, we studied DEA’s effect on Opa1 release to the cytosol, and added the data to Figure 6. Also, we modified the Results, Discussion and Materials and Methods sections accordingly.

Round 2

Reviewer 2 Report

I

I thanks the authors for response, and for the tentative to improve the paper. I am sorry that they cannot add other experiments required, because they would have added interesting information to add to those already present in the literature.

  • General consideration

My comment was referred in particular on metabolic heterogenety of different cancer cells, indeed if the authors have not the possibility to perform experiments on others/different cancer cell lines i suggest to specify in the title the use of B16F10 cells.

- Please clarify the figure 6.

Phosphorylation state of Akt and Bad were assessed in the whole -cell homogenate (indicate in the figure)

For Cyt c, Opa1 and Aif, to support the release of these proteins in the cytosol, their amount in the different fractions must be shown, the purity of fractions must be highlighted by accompanying these westerns to those of protein markers of the different fractions. I expect an increase in actin in the cytosol as well as an increase in histone in the nuclear fraction compared to whole cell homogenate.

Author Response

We thank the reviewer for raising our attention to the confusing legend of Figure 6. We modified the figure, and the text as follows:

We made the following modifications using Microsoft Word’s track changes function on the manuscript downloaded from https://susy.mdpi.com/user/manuscripts/resubmit/45d7bf9de30d69d80d6dd8785b064239

Line 235: After “prepared”, we inserted “whole-cell homogenate, and in parallel”

Line 236: We replaced “,” to “and”, and deleted “and mitochondrial”.

Line 237: We replaced “subcellular fractions” to “m”

Lines 247-248: We replaced Figure 6 to its revised version.

Line 252: After “homogenized”, we inserted “(Whole-cell). ”, and replaced “, and; finally” to Alternatively.

Line 253: We replaced “homogenates” to “harvested cells”.

Line 255: We replaced “,” to “and”; after “Opa1”, we inserted “were”.

Line 256: We replaced “and Aif” to “determined”, and replaced “and” to “, whereasthat of Aif was measured in the”.

Line 257: After “fraction”, we deleted “s, respectively, were measured”.

Line 258: We replaced “and” to “glyceraldehyde-3-phosphate dehydrogenase (GAPDH), in”.

Lines 259-261: After “fraction”, we deleted “,”; replaced “, while in” to “, ”; replaced “was used as a loading control, whereas” to “whereas in the nuclear fraction ”; replaced “a” to “loading”; and deleted “for the nuclear fraction histone H1 (HH1) were used as a loading control..”.

Line 448: In front of “and anti-actin”, we inserted “anti-GAPDH (1:2000, clone 6C5)”.

Lines 449-450: After “(Beverly, MA, USA)”, we inserted “but GAPDH that was from EMD Millipore Bioscience (Darmstadt, Germany).”.

Line 614: Below “FCS”, we inserted “GAPDH   Glyceraldehyde-3-phosphate dehydrogenase”.